# Pharmacogenomics: A Step forward Precision Medicine in Childhood Asthma

**DOI:** 10.3390/genes13040599

**Published:** 2022-03-28

**Authors:** Giuliana Ferrante, Salvatore Fasola, Velia Malizia, Amelia Licari, Giovanna Cilluffo, Giorgio Piacentini, Stefania La Grutta

**Affiliations:** 1Department of Surgical Sciences, Dentistry, Gynecology and Pediatrics, Pediatric Division, University of Verona, 37134 Verona, Italy; giorgio.piacentini@univr.it; 2Institute of Translational Pharmacology, National Research Council, 90146 Palermo, Italy; salvatore.fasola@community.unipa.it (S.F.); velia.malizia@ift.cnr.it (V.M.); stefania.lagrutta@ift.cnr.it (S.L.G.); 3Department of Pediatrics, Fondazione IRCCS Policlinico San Matteo, University of Pavia, 27100 Pavia, Italy; amelia.licari@unipv.it; 4Department of Earth and Marine Sciences, University of Palermo, 90123 Palermo, Italy; giovanna.cilluffo@unipa.it

**Keywords:** asthma, β2-agonists, children, corticosteroids, genetics, leukotriene antagonists, pharmacogenomics, treatment

## Abstract

Personalized medicine, an approach to care in which individual characteristics are used for targeting interventions and maximizing health outcomes, is rapidly becoming a reality for many diseases. Childhood asthma is a heterogeneous disease and many children have uncontrolled symptoms. Therefore, an individualized approach is needed for improving asthma outcomes in children. The rapidly evolving fields of genomics and pharmacogenomics may provide a way to achieve asthma control and reduce future risks in children with asthma. In particular, pharmacogenomics can provide tools for identifying novel molecular mechanisms and biomarkers to guide treatment. Emergent high-throughput technologies, along with patient pheno-endotypization, will increase our knowledge of several molecular mechanisms involved in asthma pathophysiology and contribute to selecting and stratifying appropriate treatment for each patient.

## 1. Introduction

Asthma is a common chronic disease in childhood, which affects about 10% of children worldwide [1]. Childhood asthma is a heterogeneous disease with respect to clinical features, severity, and response to treatment, which may reflect heterogeneity in causal pathways [2,3]. The main goal of asthma treatment is achieving and maintaining symptom control and reducing the risk of exacerbations [4]. Although most children achieve a good level of control with conventional therapies, such as inhaled corticosteroids (ICS) and/or one or more controllers, asthma still imposes a high burden, especially in children with uncontrolled symptoms [5]. It has been suggested that the inter-individual variation in drug response in asthmatic patients could be partly genetically determined [6]. Moreover, there is evidence that individuals from different populations and ethnic groups respond differently to asthma medications, likely due to genetic variants inherited from a specific ancestry associated with disease severity or response to treatment [7]. Therefore, a tailored approach is essential for improving asthma outcomes in children.

Personalized medicine, an approach to care in which individual characteristics are used for targeting interventions and maximizing health outcomes, is rapidly becoming a reality for many diseases. Its main aim is to tailor individual care by identifying clinical, biological, or genetic markers [8]. This would allow the development of diagnostic, prognostic, and therapeutic interventions specific to each patient’s needs [9]. Within the context of precision medicine, the relevant advancements obtained in the field of “omics” technologies have provided a unique opportunity for increasing the comprehension of underlying biological profiles in healthy children and the pathophysiology of multifactorial diseases [10,11]. Pharmacogenomics is a subfield of genomics, i.e., the genome-wide study of variants in the deoxyribonucleic acid (DNA), which evaluates the effect of genetic variants on the individual’s response to treatment [12]. In other words, pharmacogenomics studies the interaction between genetic variation and drug responses. Such genetic variation encompasses nucleotide repeats, deletions, insertions, and mutations that influence gene expression and/or function. Indeed, genetic variants can contribute to change the function or the number of proteins (e.g., enzymes, receptors, ion channels), and, consequently, the drug response. In addition, many intergenic and intronic variants have been proven to play a role [13].

Recent innovations in genomics technology, including phenome-wide association studies (PheWAS), whole-exome sequencing, and whole-genome sequencing (WGS), open the way to study the association between treatment response and genetic variants, taking advantage of advanced data analysis approaches based on experience called Machine Learning [14,15,16]. Most pharmacogenomic studies have been carried out in adult populations. Nonetheless, the metabolic system is still under development in children [17]. Hence, drugs may act differently in children with respect to adults, and the contribution to treatment response heterogeneity may vary at different ages.

Herein, we reviewed the latest developments related to childhood asthma in the fields of pharmacogenomics with regard to treatment response, describing its contribution to our current understanding of disease endotypes, and discuss challenges in integrating this approach into clinical practice and opportunities for future research.

## 2. Assessing Response to Treatment in Childhood Asthma: The Contribution of Pharmacogenomics

The rapidly evolving fields of genomics and pharmacogenomics may help obtain asthma control and reduce future risks in children with asthma. In particular, pharmacogenomics can detect novel molecular mechanisms and biomarkers to guide treatment. Although asthma treatment is effective in many patients, large variability in symptom control or lung function is generally observed. In the past years, several genetic variants have been associated with treatment response in asthmatic patients (Table 1).

### 2.1. Short-Acting β Agonists

Two studies focused on the change in lung function due to Short-acting β-agonists (SABA) administration, which are one of the most common reliever medications used in childhood. Mak et al. performed a WGS study on 1441 minority children with asthma to investigate the genetic association with bronchodilator response (BDR). Two DNA variants were significantly associated with BDR: rs17834628 (OR: 1.67, 95% confidence interval [CI]: 1.29–2.16, *p =* 1.18 × 10^−8^) and rs35661809 (OR: 1.59, 95% CI: 1.20–2.10, *p =* 3.33 × 10^−8^). Moreover, three specific-population loci (1p13.2 and 11p14.1 in Mexicans and 19p13.2 in African-Americans) and two shared-population loci (4q13.3 and 8q22.1) were significantly associated with SABA response as well [18]. Spear et al. conducted a genome-wide association study (GWAS) of BDR in 949 African-Americans with asthma from the Study of African Americans, Asthma, Genes and Environments. The authors identified a genetic variant on chromosome 9q21 that was significantly associated with BDR (rs73650726, *p =* 7.69 × 10^−9^). Additionally, a trans-ethnic meta-analysis across African-Americans and Latinos described three DNA variants within the intron of *PRKG1* significantly associated with BDR (rs7903366, rs7070958, and rs7081864, *p* ≤ 5 × 10^−8^). *PRKG1* encodes for a cyclic GMP-dependent protein kinase, which phosphorylates proteins involved in the feedback of the nitric-oxide signaling pathway, which is crucial in modulating vasodilation in response to β-agonists via β2-adrenergic receptors, suggesting that *PRKG1* could be a BDR candidate gene [19]. These studies indicate that the response to SABA may differ on a gene basis between racial/ethnic groups. More recently, a pilot study tested genetic variants’ association with BDR in 53 African-American children during an acute asthma attack, found three significantly associated DNA variants and provided information about the potential response to emergency treatment. Namely, rs912142 was associated with a decreased risk of low BDR (OR: 0.20, 95% CI: 0.02–0.92), while rs7081864 and rs7903366 were associated with a decreased risk of high BDR (OR: 0.097, 95% CI: 0.009–0.62) [20]. Moreover, the presence of Arg/Gly or Gly/Gly genotypes in position 16 of *ADRB2*, a gene located on chromosome 5 in a region genetically linked to asthma and BDR, was found to be significantly associated with worse BDR (post-BD Forced Expiratory Volume in the 1st second [FEV_1_]: 108.68–15.62% in Arg/Arg vs. 101.86–14.03% in Arg/Gly or Gly/Gly patients, *p =* 0.02) in 100 children with asthma [21].

### 2.2. Long-Acting β Agonists

Genetic variation has also been associated with response to long-acting β-2 agonists (LABA) in children. It is noteworthy that a very recent meta-analysis of GWAS performed in 1425 children and young adults with asthma (age 6–21 years) in regular treatment with LABA identified two loci (*TBX3* and *EPHA7*) previously involved in the response to SABA [22]. Interestingly, a recent systematic review including eight studies on children (n = 6051) showed that the *ADRB2* rs1042713 variant was significantly associated with response to LABA. In particular, five studies and a meta-analysis reported an increased risk of exacerbations in children having one or two A alleles (OR: 1.52, 95% CI: 1.17–1.99), highlighting the opportunity to investigate further the potential role of rs1042713 genotyping for personalized asthma treatment in children [23].

### 2.3. Inhaled Corticosteroids

ICS response in asthmatic patients has been investigated in many studies, as described in a recent systematic review [24]. The most consistent results have been reported for DNA variants in chromosomes 5 (rs10044254), 6 (rs6924808), 11 (rs1353649) and 16 (rs2388639). Other genetic variants have been associated with response to ICS, although not reaching genome-wide significance. The most relevant findings were reported for the *FCER2* gene, encoding for a low-affinity IgE receptor (CD23). In particular, the DNA variant rs28364072 has been associated with asthma symptoms and poor lung function, and the largest effect was reported with the risk of exacerbations (hazard ratio: 3.95, 95% CI: 1.64–9.51). However, genetic variants of two other genes (*GLCCI1*, *CRHR1*) have been successfully replicated in at least one independent study [24]. Interestingly, in a study on 263 Chinese children with asthma, *GLCCI1* rs37969 minor genotypes (TT/GT) were significantly associated with less improvement in airway hyper-responsiveness (*p =* 0.028) and significant associations were found between genetic variants rs37969, rs37972 and rs37973 minor genotypes and less improvement in pulmonary function after ICS treatment for 3 months (*p =* 0.036, *p =* 0.010 and *p =* 0.003, respectively) [25]. It is noteworthy that genetic variation at position rs7216389 in the 17q21, a widely replicated asthma susceptibility locus, has been studied in over 4000 asthmatic children/young adults treated with ICS, from 13 different studies. The 17q21 locus was found to be associated with an increased risk of oral corticosteroids use (summary OR per increase in variant allele: 1.19, 95% CI: 1.04–1.36, *p =* 0.01) and asthma-related hospitalizations/emergency visits (summary OR per increase in risk allele: 1.32, 95% CI: 1.17–1.49, *p* < 0.0001) despite ICS use [26]. Other genetic variants in genes implicated in airway inflammation, remodeling and asthma development, such as *VEGFA* and *COL2A1*, have been significantly associated with the response to ICS in children with mild-to-moderate asthma. Wan et al. found that *VEGFA* rs3025039 T allele carriers had a smaller change in FEV_1_ than CC carriers (*p =* 0.040), and in *COL2A1* rs3809324, the frequency of T allele carriers was lower than that of GG carriers (*p =* 0.048); rs3025039 was also associated with changes in FEV_1_/FVC ratio (*p =* 0.016) [27]. Moreover, in a meta-analysis of two GWAS’s performed in 1347 Hispanics/Latinos and African-American children treated with ICS, a DNA variant in the intergenic region of *APOBEC3B* and *APOBEC3C* was replicated in Europeans (rs5995653, *p =* 7.52 × 10^−3^) and was also associated with a change in FEV_1_ after a six-week treatment with ICS (*p =* 4.91 × 10^−3^). *APOBEC3B* and *APOBEC3C* encode subunits of a cytidine deaminase, a protein involved in the immune response to several viruses. Additionally, replication analyses of the genomic regions previously associated with ICS response in GWAS on European and Asian adult populations showed that the genetic variant rs62081416 near *L3MBTL4-ARHGAP28* was associated with ICS response in African-admixed children (OR: 2.44, 95% CI: 1.63–3.65, *p =* 1.57 × 10^−5^), revealing common genetic markers of response to ICS between adulthood and childhood asthma [28]. A more recent study identified a potential novel locus for ICS response in European children. A GWAS performed in 166 asthma patients (including children and young adults) found the genetic variant rs1166980 from the *ROBO2* gene, which appears to be involved in the regulation of airway inflammation, to be suggestively associated with the change in FEV_1_ after a 6-week ICS treatment (OR: 7.01, 95% CI: 3.29–14.93, *p =* 4.61 × 10^−7^) [29].

Interestingly, Dahlin et al. have previously identified in a genome-wide interaction study (GWIS) of 1321 adults and children with asthma, age-by-genotype interactions in several asthma candidate genes, suggesting that age-specific genetic mechanisms may be implicated in the response to ICS as measured by the occurrence of exacerbations. In particular, the top-ranked age-by-genotype association was found for the DNA variant rs34631960 in *THSD4*, a gene potentially involved in lung function, airway remodeling, and asthma severity, which could be protective against the risk of exacerbations in younger asthmatics on ICS treatment, or, conversely, may predict an increased risk of poor ICS response in older patients [30].

Nonetheless, age is just one of the factors associated with treatment response in asthma, as suggested by some studies showing that subjects from different ethnic groups respond differently to medications such as ICS and LABA [34,35,36]. Pharmacogenomic studies in asthma primarily involved White people of European descent. A GWAS tested in 2681 children of European descent on ICS treatment found associations with 10 independent variants. Among them, one variant at the *CACNA2D3-WNT5A* locus, which could have a role in asthma, was replicated in Europeans (rs67026078; *p =* 0.010), but not in non-European populations [31].

### 2.4. Inhaled Corticosteroids + Long-Acting β Agonists

A very recent study from the Best African Response to Drug trials, in which children and adolescents with uncontrolled asthma with low-dose ICS received different step-up combination therapies, identified novel genetic variation involved in differential response to treatment. The two co-primary outcome comparisons were the step-up from low-dose ICS to the quintuple dose of ICS (250 μg twice daily in children, 500 μg twice daily in adolescents and adults) vs. double dose (100 μg twice daily in children, 250 μg twice daily in adolescents and adults), and the quintuple dose of ICS vs. 100 μg fluticasone plus a LABA (salmeterol 50 μg twice daily). With respect to children, a locus for quintupling ICS vs. adding a LABA was identified close to *RNFT2* and *NOS1* (rs73399224, OR: 0.7, 95% CI: 0.07–0.42, *p =* 8.4 × 10^−5^), which have been associated with asthma risk and airway inflammation. These results highlight the need to include ancestral diversity to identify reliable profiles of drug response in asthmatic patients [32].

### 2.5. Leukotriene Modifiers

Large variability has been reported in patient response to asthma controller medications, including leukotriene modifiers (LTMs). In a recent systematic review including 26 studies on LTMs, no consistent findings were reported for candidate gene studies of LTMs. Moreover, a lack of replication of genetic variants associated with poor LTMs response was described [24]. More recently, a study investigated the role of the *LTA4H* genetic variant rs2660845 and the age of asthma onset in response to montelukast in 3594 patients (2514 late-onset: >18 years, and 1080 early-onset: ≤18 years) from seven cohorts. A meta-analysis of 523 early onset individuals from European ancestry showed an increased risk of asthma attacks in those carrying at least one G allele, despite montelukast treatment (OR: 2.92, 95% CI: 1.04–8.18, *p =* 0.0412) compared to those in the AA group. This finding suggests that the genetic variation in *LTA4H*, together with the age of disease onset, may explain the variability in treatment response, likely due to the up-regulated activity of *LTA4H*, resulting in high Leukotriene-B4 blood concentration, which can trigger airway inflammation and responsiveness [33].

## 3. Application of Pharmacogenomics in Childhood Asthma Practice: Challenges and Evidence Gaps

Starting from patients’ genetic information, pharmacogenomics aims to contribute to predicting responses to the given treatment. There is evidence that a pharmacogenomic approach enhances the knowledge necessary for treatment decision-making in the pediatric care setting, providing information for drug selection and dosing options [37]. However, although pharmacogenomic testing has recently mainly become available and less costly, its use in pediatric clinical practice is scarce, especially outside academic centers and specialized institutions. Major barriers to its implementation include difficulties in interpretation of test findings and the need for education and support in decision-making for pediatricians. Generally, clinicians and healthcare providers are not confident approaching pharmacogenomics in routine clinical practice [38]. Indeed, a recent survey of pediatricians revealed that, although 80% recognized that pharmacogenomic testing would improve treatment efficacy and safety, fewer than 10% were familiar with pharmacogenomics. Moreover, most were interested in educational training, particularly in test interpretation and treatment recommendations [39]. Therefore, education on pharmacogenomics through *ad hoc* training courses and e-learning modules should be increasingly delivered to healthcare professionals, including pediatricians, and should be included in the curriculum of every university medical school.

Furthermore, many other issues have to be addressed before such an innovative approach could be integrated in daily clinical practice. First, besides collecting evidence and reporting measures of association, measures of clinical validity (e.g., number needed to genotype/treat) and predictive values of utility (e.g., positive predictive value, negative predictive value) should be assessed [40].

Another important issue to be evaluated before translating pharmacogenomics from research to clinical practice is cost effectiveness. Additionally, a proper infrastructure is a key condition to implement pharmacogenomics into real life. The involvement of all stakeholders, sufficient facilities for genotyping, and the availability of patients’ genotype data in electronic healthcare records are crucial to convey pharmacogenomics to the clinic. Using point-of-care tests for DNA variants that are going to be commonly evaluated should be considered, especially when a rapid response is required [41]. As an example, pharmacogenomics could be crucial in managing acute asthma exacerbations. In this context, feasibility was recently described for a pharmacogenetic study set in the emergency department (ED) for children with acute asthma exacerbations. Quality DNA collection through buccal swabs and BDR measurement was obtained in 59 patients. According to most of the providers, the study did not influence their workflow while reporting difficulties with spirometry in young children. Thus, these findings support the potential use of point-of-care pharmacogenomic testing in the ED to improve childhood asthma treatment [42]. Nonetheless, whereas the use of genetics to guide therapeutic interventions in pediatrics is currently limited to less common diseases, such as cystic fibrosis [43], current international guidelines for asthma do not recommend it. In spite of this, international consortia have been created [44], and evidence suggests that pharmacogenomic testing should be established in childhood asthma practice. Interestingly, a pilot prospective questionnaire-based study recently conducted in children and young people with asthma, their parents, and healthcare professionals at a secondary/tertiary children’s hospital in the UK showed the acceptability of using genetic data for asthma management. In particular, 46% of participants were happy about sharing genetic data with healthcare providers, and 46% agreed to share solely to guide asthma management [45]. Further studies are required to develop and implement successful pharmacogenomic services in pediatric daily clinical practice.

Another potential reason for the poor application of the current knowledge on pharmacogenomics in the clinical management of asthma patients is the little functional evidence and the lack of experimental studies to understand how the genes identified could be involved in the response to asthma medications. Therefore, studies using cell cultures and animal models should be carried out in order to establish a link between genetic markers and biological pathways.

Additionally, further validation of the genes identified to date, as well as investigation attempting to improve the statistical power of the studies carried out until now and to identify loci accounting for a larger proportion of the variation in treatment response should also be considered.

The main challenges and evidence gaps associated with pharmacogenomics in childhood asthma research and practice are summarized in Figure 1.

## 4. Future Research Perspectives

Despite the numerous genetic variants identified, the poor replication of the results obtained in independent study populations only partly explains heterogeneity to treatment response in childhood asthma. Indeed, replication of genetic associations is also complicated by the heterogeneity in outcome measures as well as in study populations. In addition, most studies are underpowered, effect sizes are often limited, and large sample sizes would be required for identifying significant effects [17,46].

International collaboration may therefore be useful to identify genetic markers in large sample sizes of well-phenotyped asthmatic children. Looking at this, the Pharmacogenomics in Childhood Asthma (PiCA) consortium was established as the first consortium focusing on pharmacogenomics in childhood asthma. PiCA’s main goals are creating a platform to discover new pharmacogenomic markers by means of GWAS meta-analyses, replicating identified loci associated with treatment response, and finally developing algorithms to guide asthma therapy [44].

A relevant issue to address in pharmacogenomics is the identification of genetic markers associated with treatment response in patients with different ethnicities to guide asthma treatment regardless of the ethnic background. Moreover, combining DNA variants in risk scores might be helpful in explaining the observed treatment heterogeneity [46]. Moreover, translational research is needed, beyond studies conducted in highly controlled settings [47]. Another important goal is to link loci identified in GWAS to biological pathways of the disease [48].

Given the existence of many asthma pheno/endotypes, a promising approach in asthma pharmacogenomics is through applying PheWAS, which tests for associations between genetic variants and a wide range of phenotypes in a given population [49]. In a recent meta-PheWAS, a novel association of the *IFIH1* loss-of-function allele rs1990760-C with risk of asthma was identified [50]. Therefore, PheWAS could be an innovative tool to determine the pharmacogenomics in different asthma phenotypes and could be considered complementary to GWAS [51]. Indeed, unlike GWAS, which examines the association of the genotype with a specific phenotype, PheWAS determines which phenotype is associated with a given genotype. It is noteworthy that asthma phenotypes and response to treatment cannot be explained by genomic susceptibility alone. Instead, the interaction between genetic background and the environment might have a major role, and GWIS might provide novel insights into these complex relationships. Nonetheless, the application of pharmacogenomics in childhood asthma should also take into account the heterogeneous disease endotypes. The potential of pharmacogenomics to improve asthma treatment and to correlate endotypes with therapeutic response is considerable. So far, pharmacogenomics studies have focused on candidate genes and have identified genetic loci associated with responsiveness to ICS, β2-agonists and LTMs, whereas a tailored approach would be necessary to improve the response to biological drugs that are directed to a specific biologic pathway [52]. Indeed, approaching the interdependencies among genetics, environment and biologic pathways that define the different asthma pheno/endotypes would allow the identification of reliable precision medicine profiles of drug response in asthmatic children. Therefore, future studies should include a multitude of biomarkers, both genetic and non-genetic ones, for assessing which combination may be the most useful in clinical practice. Actually, one of the main challenges for the implementation of pharmacogenomics in clinical practice is the lack of validated and useful biomarkers. High-throughput technologies hold the promise of expanding our knowledge of molecular mechanisms underpinning asthma pathophysiology and may contribute to select and stratify targeted therapeutic strategies. In particular, the rapid advances in bioinformatics may allow approaching different -omics layers simultaneously, given that many studies have shown associations between different -omics markers, such as epigenomics, transcriptomics, breathomics, and asthma-related outcomes [53,54,55]. Nevertheless, integrating multi-omics and clinical data needs large-scale databases, strong computational power, and close collaboration between clinicians with expertise in asthma and bioinformaticians [46]. Recently, pan-European research infrastructures provided new services for the community of scientists. In particular, the Biobanking and BioMolecular resources (BBMRI-ERIC) infrastructure [56], sustains the collection of biological samples, such as blood, tissues or DNA that may be useful to detect new targets for therapy and may support drug discovery and development. In addition, the ELIXIR infrastructure [57], which integrates and sustains bioinformatics resources across European life science organizations, may provide new insights from large data sets, particularly data from gene sequencers.

Finally, developing pediatricians’ competency in genomics is another relevant objective to accomplish in the future. Genomics will provide pediatricians with new insights into the biological basis of childhood asthma. Therefore, future research should consider the educational needs of pediatricians and evaluate strategies and resources tailored to the peculiarity of their specialty [58].

## 5. Conclusions

In summary, though pharmacogenomics studies have provided insight into the genetic mechanisms implicated in asthma treatment response, more research is needed to obtain predictive markers with clinical relevance. Though many genetic variants have been shown to influence response to short- and long-acting β-2 agonists, inhaled corticosteroids, and leukotriene modifiers, results are still inconsistent and/or effect sizes are small. Furthermore, considering that different proteins could be involved in the treatment response, the individual effect of one genetic variant is limited. It should also be considered that epigenetic changes, gene–gene and gene–environment interactions could affect pharmacogenomic associations [6].

Challenges may be even more significant for pediatric patients transitioning to adult care. In this context, longitudinal data collection would allow evaluating long-term outcomes and the overall clinical utility of pharmacogenomic testing in daily asthma practice [56]. Integrating multi-omics and clinical data might improve the ability to predict treatment response in children with asthma and to build decision support tools potentially useful in the selection of drugs, in particular emergent and expensive biologicals, and to predict adverse events as well as exacerbations and decline in lung function.

## Figures and Tables

**Figure 1 genes-13-00599-f001:**
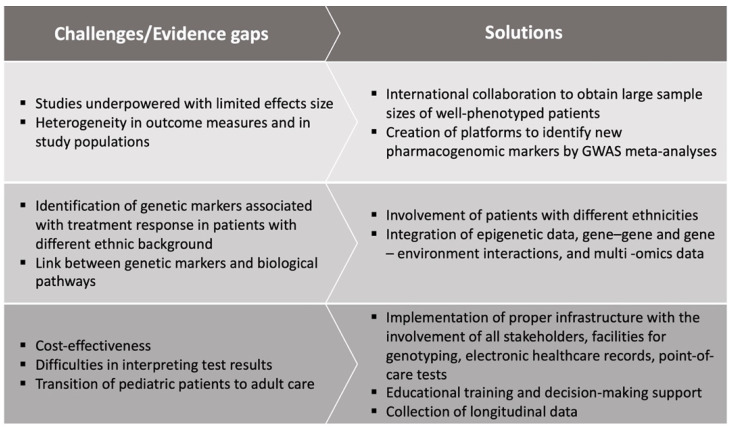
Challenges and evidence gaps associated with pharmacogenomics in childhood asthma research and practice.

**Table 1 genes-13-00599-t001:** Summary of genetic variants associated with asthma treatment response in children and adolescents.

Author, Year	Study Design	Study Population	Aim	Results	Comments
**Short-acting β agonists**					
Mak, A.C.Y.; et al. Am J Respir Crit Care Med. 2018 [18]	WGS study	1441 minority children with asthma from three ethnic groups (483 Puerto Ricans, 483 Mexicans, 475 African-Americans)	To identify genetic association with BDR	rs17834628 (OR: 1.67, 95% CI: 1.29–2.16, *p =* 1.18 × 10^−8^) and rs35661809 (OR: 1.59, 95% CI: 1.20–2.10, *p =* 3.33 × 10^−8^), including genome-wide significant (*p* < 3.53 × 10^−7^) and suggestive (*p* < 7.06 × 10^−6^) three specific-population loci (1p13.2 and 11p14.1 in Mexicans and 19p13.2 in African-Americans) and two shared-population loci (4q13.3 and 8q22.1) near genes *DNAH5*, *NFKB1*, *PLCB1*, *ADAMTS3* and *COX18* significantly associated with BDR	Population-specific and shared genetic variants were associated with BDR in three different ethnic populations of children with asthma
Spear M.L.; et al. JAMA Pharmacogenomics J. 2019 [19]	GWAS	949 African American minority children with asthma	To identify genetic association with BDR	rs73650726 on chromosome 9q21 (*p =* 7.69 × 10^−9^) and three DNA variants within the intron of *PRKG1* significantly associated with BDR (rs7903366, rs7070958, and rs7081864, *p* ≤ 5 × 10^−8^)	Population-specific and shared DNA variants contribute to differences in BDR in minority children with asthma
Fishe, J.N.; et al. Pharmacogenet Genomics. 2021 [20]	Prospective pilot study	53 African American children with asthma	To identify genetic association with BDR during an acute asthma attack	rs912142 associated with decreased risk of low BDR (OR: 0.20; 95% CI: 0.02–0.92); rs7081864 and rs7903366 associated with decreased risk of high BDR (OR: 0.097; 95% CI: 0.009–0.62)	Genetic variants provide information regarding a child’s potential response to emergency asthma exacerbation treatment
Scaparrotta, A.; et al. J Aerosol Med Pulm Drug Deliv. 2019 [21]	Preliminary observational prospective study	100 children with persistent asthma	To verify the association of genetic variants of *ADRB2*, *THRB* and *ARG1* with BDR	Arg/Gly or Gly/Gly rs1042713 significantly associated with a worse BDR (post-BD FEV_1_: 108.68–15.62% in Arg/Arg vs. 101.86–14.03% in Arg/Gly or Gly/Gly patients, *p =* 0.02). No significant association for the other three examined DNA variants	Arg16Gly in *ADRB2* gene is associated with a worse response to SABA in children with persistent asthma
**Long-acting β agonists**					
Slob, E.M.A.; et al. Pediatr Allergy Immunol. 2021 [22]	Meta-analysis of GWAS	1425 children and young adults with asthma	To identify genetic association with exacerbations	Eight genetic variants suggestively (p threshold ≤5 × 10^−6^) associated with exacerbations despite LABA use; two DNA variants near loci *TBX3* and *EPHA7* identified.	Pharmacogenetic markers can determine whether children experience exacerbations despite LABA use
Slob, E.M.A.; et al. Pediatr Allergy Immunol. 2018 [23]	Systematic review of pharmacogenetic studies in patients with asthma treated with LABA	6051 children with asthma	To identify genetic variants associated with LABA response	The *ADRB2* rs1042713 variant was significantly associated with response to LABA; increased risk of exacerbations found in children carrying one or two A alleles (OR: 1.52, 95% CI: 1.17–1.99)	*ADRB2* rs1042713 variant is associated with response to LABA in children
**Inhaled corticosteroids**					
Farzan, N.; et al. Clin Exp Allergy J. 2017 [24]	Systematic review of pharmacogenomics and pharmacogenetics of ICS in patients with asthma	Children and adolescents from 29 candidate gene studies and 4 GWAS	To identify genetic variants associated with ICS response	The *FCER2* rs28364072 variant associated with asthma symptoms, poor lung function, the largest effect reported with the risk of exacerbations (hazard ratio: 3.95; 95% CI: 1.64–9.51)	A lack of replication of genetic variants is associated with poor ICS response. Most consistent findings found for the *FCER2* gene
Huang, J.; et al. BMC Pulm. Med. 2020 [25]	Observational prospective study	263 children with asthma	To determine the associations between *GLCCI1* genetic variants and ICS response	*GLCCI1* rs37969 minor genotypes (TT/GT) associated with less improvement in airway hyper-responsiveness (*p =* 0.028); significant associations between genetic variants rs37969, rs37972 and rs37973 minor genotypes and less improvement in pulmonary function (*p =* 0.036, *p =* 0.010 and *p =* 0.003, respectively)	*GLCCI1* rs37969 minor genotypes is associated with less improvement in airway hyper-responsiveness. *GLCCI1* rs37969, rs37972 and rs37973 genetic variants are associated with pulmonary function in children with asthma after ICS treatment.
Farzan, N.; Allergy. 2018 [26]	Observational study	4000 asthmatic children/young adults treated with ICS	To study the association between genetic variant 17q21 rs7216389 and asthma exacerbations despite ICS use	17q21 rs7216389 associated with an increased risk of oral corticosteroids use (summary OR per increase in variant allele: 1.19, 95% CI: 1.04–1.36, *p =* 0.01) and asthma-related hospitalizations/emergency visits (summary OR per increase in risk allele: 1.32, 95% CI: 1.17–1.49, *p* < 0.0001)	17q21 is associated with an increased risk of exacerbations in children/young adults treated with ICS
Wan, Z.; et al. Pharmacogenomics. 2019 [27]	Observational prospective study	128 children with mild-to-moderate asthma	To investigate the involvement of genetic variants in *VEGFA*, *TBX21* and *COL2A1* in the response to ICS	Change in FEV_1_ after ICS treatment in *VEGFA* rs3025039 minor homozygotes (TT) and heterozygotes (CT) smaller than that in major homozygotes (CC) (*p =* 0.040), and associated with changes in FEV_1_/FVC ratio (*p =* 0.016). Children with the minor homozygous (TT) and heterozygous (GT) genotypes at *COL2A1* rs3809324 had less improvement in FEV_1_ than those with the major homozygous (GG) genotype (*p =* 0.048)	*VEGFA* and *COL2A1* variants are associated with the response to ICS in asthmatic children
Hernandez-Pacheco, N.; et al. Clin Exp Allergy J. 2019 [28]	Meta-analysis of GWAS	1347 Hispanics/Latinos and African-American children with asthma on ICS treatment	To identify genetic variants associated with asthma attacks in children on ICS treatment, and to validate previous GWAS findings	DNA variant rs5995653 in the intergenic region of *APOBEC3B* and *APOBEC3C* replicated in Europeans (*p =* 7.52 × 10^−3^) and associated with change in FEV_1_ (*p =* 4.91 × 10^−3^). DNA variant rs62081416 near *L3MBTL4-ARHGAP28* associated with ICS response in African-admixed children (OR: 2.44, 95% CI: 1.63–3.65, *p =* 1.57 × 10^−5^)	*APOBEC3B* and *APOBEC3C* genes are associated with ICS response in asthmatic children. The association of the *L3MBTL4-ARHGAP28* genomic region previously described in a GWAS of ICS response in subjects of European descent was validated in admixed children
Hernandez-Pacheco, N.; et al. J Pers Med. 2021 [29]	GWAS	166 asthma patients, including children and young adults	To identify novel genetic variants involved in ICS response in patients with asthma	The DNA variant rs1166980 from the *ROBO2* gene associated with change in FEV_1_ (OR: 7.01, 95% CI: 3.29–14.93, *p =* 4.61 × 10^−7^)	*ROBO2* is a potential novel locus for ICS response in Europeans
Dahlin, A.; Plos One. 2020 [30]	GWIS	1321 patients with asthma, including children and adults	To identify genetic variants associated with response to ICS	The top-ranked age-by-genotype association found for the DNA variant rs34631960 in *THSD4*, which could be protective against exacerbations risk in younger patients taking ICS, or may predict an increased risk of poor ICS response in older patients	Age-specific genetic mechanisms may regulate response to ICS
Hernandez-Pacheco, N.; et al. Eur Respir J. 2021 [31]	GWAS	2681 children with asthma of European descent on ICS treatment	To identify genetic variants associated with asthma exacerbations	Ten genetic variants associated with asthma exacerbations (*p* ≤ 5 × 10^−6^). One variant at the *CACNA2D3-WNT5A* locus replicated in Europeans (rs67026078; *p =* 0.010), but not in non-European populations.	The intergenic region of *CACNA2D3* and *WNT5A* could be a novel locus for asthma exacerbations despite ICS treatment in European children
**Inhaled corticosteroids + Long-acting β agonists**					
Ortega, V.E.; et al. Lancet Child Adolesc Health. 2021 [32]	Ancestry-based pharmacogenetic studies of children, adolescents and adults from the Best African Response to Drug trials	249 children and 267 adolescents and adults of African descent	To understand the pharmacogenetic mechanisms regulating therapeutic responsiveness to ICS + LABA	In children, a locus for quintupling ICS vs. adding a LABA identified close to *RNFT2* and *NOS1* (rs73399224, OR: 0.7, 95% CI: 0.07–0.42, *p =* 8.4 × 10^−5^)	Including ancestral diversity is crucial in the identification of reliable precision medicine profiles of drug response in asthmatic patients
**Leukotriene modifiers**					
Farzan, N.; et al. Clin Exp Allergy J. 2017 [24]	Systematic review of pharmacogenomics and pharmacogenetics of LTMs in patients with asthma	Children and adolescents from 24 candidate gene studies and 2 GWAS	To identify genetic variants associated with LTMs response	No consistent findings for candidate gene studies of LTMs.	A lack of replication of genetic variants is associated with poor LTMs response
Maroteau, C.; Plos One. 2021 [33]	Meta-analysis of seven cohort studies	3594 patients with asthma treated with montelukast for at least 6 months (2514 late-onset: >18 years, and 1080 early-onset: ≤18 years) from seven cohorts	To investigate the role of the *LTA4H* genetic variant rs2660845 and the age of asthma onset in response to montelukast	Increased risk of exacerbation under montelukast treatment in European individuals with early-onset carrying at least one copy of rs2660845 (OR: 2.92, 95% CI: 1.04–8.18, *p =* 0.0412)	Genetic variation in *LTA4H*, together with the age of asthma onset, may contribute to variability in montelukast response

BDR: bronchodilator response; CI: confidence interval; FEV_1_: Forced Expiratory Volume in the 1st second; GWAS: genome-wide association study; GWIS: genome-wide interaction study; ICS: Inhaled corticosteroids; LABA: Long-acting β agonists; LTMs: Leukotriene modifiers; OR: odd ratio; SABA: Short-acting β agonists; WGS: Whole-genome sequencing.

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
