# Peer review of "Pharmacogenomics: A Step forward Precision Medicine in Childhood Asthma"

_genes, 2022, doi:10.3390/genes13040599_

Round 1
Reviewer 1 Report
The authors have well described the main findings of the asthma pharmacogenomic studies carried out in the last years in this review, also pointing out the major limitations and challenges of the investigation conducted to date and the directions that future studies should follow. Nonetheless, I have some comments the authors might want to consider:
Major comments
- Page 1, lines 37-39: Substantial differences in the response to asthma medications among populations and ethnic groups have also been reported. A remark about this matter should be added.
- Page 2, lines 53-55: The mechanisms through with SNPs may be involved in the treatment response is not only restricted to changes in the translated protein. Indeed, many intergenic and intronic variants have been evidenced to play a role here. In addition to other types of genetic variation. The authors should take this into account.
- Please, use italic font for all the human genes mentioned along with the manuscript.
- Page 7, line 198: Although this reviewer strongly agrees with the statement encouraging the education of healthcare professions into pharmacogenomics and the modification of the current infrastructure to be able to introduce the findings of the investigations conducted so far into the clinical practice, this reviewer does not believe these are the only limitations and potential reasons of the scarce or absent application of the current knowledge.
One of the major concerns is the little functional evidence and experimental studies to understand how the genes identified could be involved in the response to asthma medications. For that, big efforts on investigating this through cell culture and animal models should become a reality before conceiving the application of the loci associated in the clinical management of asthma patients in the near future.
Additionally, further validation of the genes identified to date as well as investigation attempting to overcome the limitations of the studies carried out until now to improve the statistical power to identify loci that accounts for a larger proportion of the variation in treatment response should also be taken into account. Although this has been summarized in Figure 1, I would like to encourage the authors to include these considerations in the manuscript in further detail.
Minor comments
- Table 1: I would like to recommend the authors try to simplify this table and reduce the amount of text included. It seems a bit overwhelming and unclear. I would replace the “Aim” with a “Treatment” column, for instance.
- In this reviewer’s opinion, it would be clearer and easier to follow if the description of the pharmacogenomic studies carried out in the last years would be separated into different sections based on the asthma treatment evaluated, for example.
- Page 3, line 119: Including the reference of that previous review might be helpful for the reader.
- Page 3, line 131: The definition of ICS response would be appreciated.
Reviewer 2 Report
The authors present a review manuscript on pharmacogenomics in childhood asthma. The manuscript presents an overview of the latest GWAS in the field and discusses the challenges and future directions that need to be addressed before pharmacogenomics will be implemented in clinical practice.
The authors must address some points listed below:
- The authors should add the latest findings of the candidate-gene studies as well. GWAS identifies novel candidate genes, while candidate-gene studies are needed to replicate/validate/confirm the results of GWAS. The replication of genes/genetic variants is crucial for the implementation of pharmacogenomics in clinical practice.
- Phenome-wide association studies are a novel approach, and since many readers are not yet familiar with this, it should be explained. This is a very good approach to study asthma (pharmaco)genomics since the existence of several phenotypes/endotypes under the asthma umbrella are well documented. The asthma pharmacogenomics should follow this aspect as well: determine the pharmacogenomics in several or different asthma phenotypes. And in this context, the PheWAS represent an ideal tool. The authors should include the usefulness of PheWAS in asthma pharmacogenomics. And include this in the paragraph “4. Future research perspectives”. Main future directions should be novel tools (PheWAS) and replication of results in independent populations. The most important challenge for the implementation of pharmacogenomics in clinical practice is still the lack of validated and useful biomarkers.
- In the same context, the influence of asthma endotypes on the (results) of pharmacogenomics should be discussed in greater detail.
Minor:
- Use genetic variants or DNA variants instead of single nucleotide polymorphisms (SNPs).
Round 2
Reviewer 1 Report
The authors have appropriately addressed all the reviewers' suggestions and comments.
Reviewer 2 Report
The authors have addressed all the questions, and the manuscript is improved and acceptable for publication.
I have just a minor point:
The authors should use minor/major genotypes/homozygotes,… instead of mutant/wild-type.